# Splicing of enhancer-associated lincRNAs contributes to enhancer activity

Jennifer Y Tan[1], Adriano Biasini[1], Robert S Young[2], Ana C Marques[1]

**Transcription is common at active mammalian enhancers sometimes giving rise to stable enhancer-associated long intergenic noncoding RNAs (elincRNAs). Expression of elincRNA is associated with changes in neighboring gene product abundance and local chromosomal topology, suggesting that transcription at these loci contributes to gene expression regulation in *cis*. Despite the lack of evidence supporting sequence-dependent functions for most elincRNAs, splicing of these transcripts is unexpectedly common. Whether elincRNA splicing is a mere consequence of cognate enhancer activity or if it directly impacts enhancer function remains unresolved. Here, we investigate the association between elincRNA splicing and enhancer activity in mouse embryonic stem cells. We show that multi-exonic elincRNAs are enriched at conserved enhancers, and the efficient processing of elincRNAs is strongly associated with their cognate enhancer activity. This association is supported by their enrichment in enhancer-specific chromatin signatures; elevated binding of co-transcriptional regulators; increased local intra-chromosomal DNA contacts; and strengthened *cis*-regulation on target gene expression. Our results support the role of efficient RNA processing of enhancer-associated transcripts to cognate enhancer activity.**

## Introduction

Enhancers are distal DNA elements that positively drive target gene expression (Banerji et al, 1981; Moreau et al, 1981; Li et al, 2016). These regulatory regions are DNase I hypersensitive, marked by histone 3 acetylation at lysine 27 (H3K27ac), and a high ratio of monomethylation versus trimethylation at histone 3 lysine 4 (H3K4me1 and H3K4me3, respectively). Together, these chromatin signatures are commonly used to annotate enhancers genome wide (Hoffman et al, 2012). Most active enhancers are also transcribed (De Santa et al, 2010; Kim et al, 2010; Kowalczyk et al, 2012). Relative to non-transcribed enhancers, those that give rise to enhancer-associated transcripts are more strongly associated with enhancer-specific chromatin signatures (Wang et al, 2011) and display higher levels of reporter activity both in vitro (Wu et al, 2014; Young et al, 2017) and in vivo (Andersson et al, 2014), supporting the link between enhancer transcription and *cis*-regulatory function. Whereas most enhancers transcribe short noncoding RNAs that are non-polyadenylated, unspliced, and short-lived from both the sense and antisense strands (eRNAs) (Kim et al, 2010), a subset of enhancers are predominantly transcribed in one direction (Natoli & Andrau, 2012) and produce enhancer-associated long intergenic noncoding transcripts that we refer to as elincRNAs (Marques et al, 2013). The asymmetry of transcriptional activity at these enhancers is at least in part due to differences in transcript stability. Specifically, and in contrast to eRNAs, elincRNAs are polyadenylated, relatively long, stable, and frequently spliced (Koch et al, 2011; Marques et al, 2013; Hon et al, 2017).

Enhancer transcription can increase local chromatin accessibility (Mousavi et al, 2013), modulate chromosomal interactions between cognate enhancer and target promoters (Lai et al, 2013), and regulate the load, pause, and release of RNA Polymerase II (RNAPII) (Maruyama et al, 2014; Schaukowitch et al, 2014), ultimately contributing to enhanced expression of neighboring protein-coding genes (Orom et al, 2010; Marques et al, 2013). Recently, we showed that elincRNAs preferentially locate at topologically associating domain (TAD) boundaries, and their expression correlates with changes in local chromosomal architecture (Tan et al, 2017). Although the association between elincRNA transcription and enhancer activity is relatively well established, whether the molecular mechanisms underlying their functions depend on their transcript sequences has not yet been unequivocally demonstrated. Notably, consistent with the absence of nucleotide conservation at their exons (Marques et al, 2013), many elincRNA functions appear to rely on transcription alone (Yoo et al, 2012; Lai et al, 2013; Li et al, 2013; Hsieh et al, 2014; Alexanian et al, 2017).

Despite evidence that the functions of most elincRNAs is likely transcription dependent, a relatively large proportion of elincRNAs is not only stably transcribed but also undergoes splicing (Marques et al, 2013; Hon et al, 2017; Krchnakova et al, 2019). Recently, splicing of Blustr, a lincRNA expressed in mouse embryonic stem cells (mESCs) whose transcriptional start site initiates from an active enhancer (Mouse ENCODE Consortium et al, 2012), was shown to be sufficient to modulate the expression of its

[1]Department of Computational Biology, University of Lausanne, Lausanne, Switzerland  [2]Medical Research Council Human Genetics Unit, Medical Research Council Institute of Genetics & Molecular Medicine, University of Edinburgh, Edinburgh, UK

Correspondence: anaclaudia.marques@unil.ch; jennifer.tan@unil.ch

cognate protein-coding gene target in *cis* (Engreitz et al, 2016). Removal of the splicing signals in another elincRNA, Haunt, by replacing its endogenous locus with its cDNA, could not rescue its *cis*-regulatory function (Yin et al, 2015). Recently, the genome-wide analysis of enhancer transcription across multiple human cells lines (Gil & Ulitsky, 2018) corroborates candidate loci analyses, supporting the association between elincRNA splicing and cognate enhancer activity.

Here, we investigate the association between elincRNA splicing and developmentally regulated mESC enhancer's activity. We show that efficient splicing of multi-exonic elincRNAs associates with higher activity, cell-type–specific function and increased conservation of their cognate enhancers.

# Results

To annotate enhancer-associated lincRNAs (elincRNAs), we took advantage of the extensive publicly available data for transcription and chromatin signatures in pluripotent mESC. We considered all intergenic mESC enhancers overlapping a DNase I–hypersensitive region (The ENCODE Project Consortium, 2012) and annotated their associated transcripts using a stringent approach that required the overlap between their transcriptional start site and the enhancer. This led to the identification of a relatively small, yet high confidence, set of enhancer-associated lincRNAs (n = 100, elincRNAs, Table S1) and eRNAs (n = 2,117). As expected (Xu et al, 2009; Andersson et al, 2014; Young et al, 2017), we found divergent transcription at all promoter and enhancer-associated transcriptional initiation regions (TIRs, Fig 1A–D). In contrast to eRNA-producing enhancers (Fig 1A), enhancers associated with elincRNAs (Fig 1B) have transcriptional profiles that resemble those of other promoter-associated mESC transcripts, including other mESC-expressed non–enhancer-associated lincRNAs (oth-lincRNAs) (Fig 1C) and protein-coding genes (Fig 1D).

Given the relatively small number of the stringently annotated elincRNAs, we also annotated elincRNAs using a less stringent approach. Analysis of this less stringent and more comprehensive set of mESC elincRNA (1,983 elincRNAs of which 211 are multi-exonic) is

described in the Supplemental Data 1 and fully supports the analysis of the stringently annotated set of mESC elincRNAs.

## Multi-exonic elincRNAs are associated with stronger enhancer activity

Next, we investigated whether elincRNA splicing is linked to its cognate enhancer activity. Given that most enhancer activity is tissue specific, we first investigated the association between enhancer transcription and putative target expression during embryonic neurogenesis (Fraser et al, 2015). Similar to what was described previously (Marques et al, 2013), we found that elincRNA transcription positively correlated with changes in neighboring protein-coding gene abundance (Fig S1A). This association is 2.5-fold stronger for multi-exonic elincRNAs (median FD target transcription = 0.49) than their single-exonic counterparts (median FD target transcription = 0.19, *P* < 0.05, two-tailed Mann–Whitney *U* test, Fig 2A). As expected, no association was observed for other transcript classes, regardless of their splicing activity (Fig S1B).

Consistent with their stronger association with neighboring protein-coding gene expression, chromatin signatures associated with high enhancer activity were found at enhancers that transcribe multi-exonic elincRNAs compared with those that give rise to either single-exonic elincRNAs or eRNAs. Specifically, multi-exonic elincRNA-producing enhancers were enriched for monomethylation of histone 3 lysine 4 (H3K4me1, Fig 2B), acetylation of histone 3 lysine 27 (H3K27ac, Fig 2C), and DNase I accessibility (DHSI, Fig 2D). Using a hypothesis-free approach, we found that relative to their unspliced counterparts, TIRs of multi-exonic elincRNAs were significantly enriched (false discovery rate < 0.05) for transcription factor–binding motifs required for the recruitment of the transcriptional co-activator cAMP-response element-binding protein (CREB)–binding protein (CREBBP) (Bedford et al, 2010), including Stat1, Egr1, Sp2, Smad3, and Klf5 (Table S2). For a subset of the enriched CREBBP-recruiting transcription factors with available chromatin immunoprecipitation (ChIP) sequencing data in mESCs and the CREBBP transcriptional co-activator, EP300 (Merika et al, 1998), we found experimental support for their more frequent binding at multi-exonic elincRNAs' TIRs (Figs 2E and S1C–E). Recently, direct binding of CREBBP to enhancer-associated RNAs was demonstrated to stimulate its histone acetylation activity and induce activation of target gene transcription (Bose et al, 2017). Our findings raise the possibility that

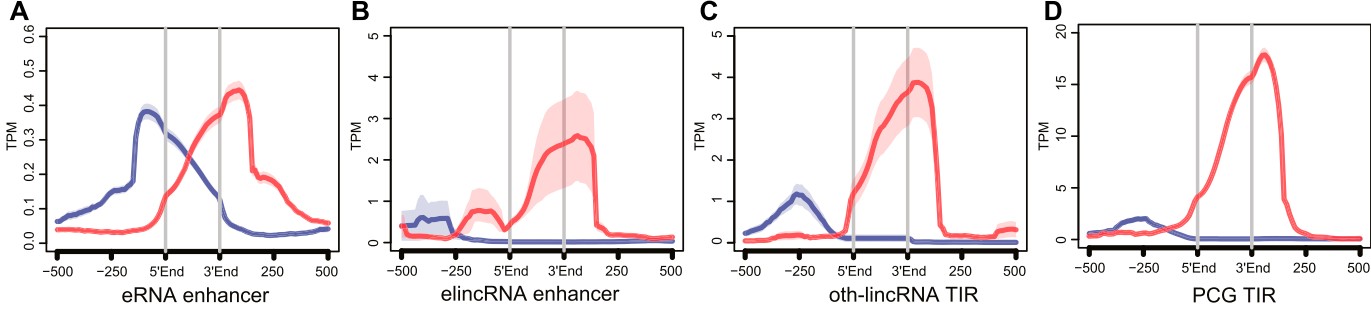

**Figure 1. Stringent annotations of elincRNAs.**
**(A, B, C, D)** Metagene plots of CAGE reads centered at transcription initiation regions (TIRs) of (A) eRNAs, (B) elincRNAs, (C) other mouse embryonic stem cell-expressed lincRNAs (oth-lincRNAs), and (D) protein-coding genes (PCGs). Sense (red) and antisense (blue) reads denote those that map to the same or opposite strand, respectively, as the direction of their cognate TIRs.

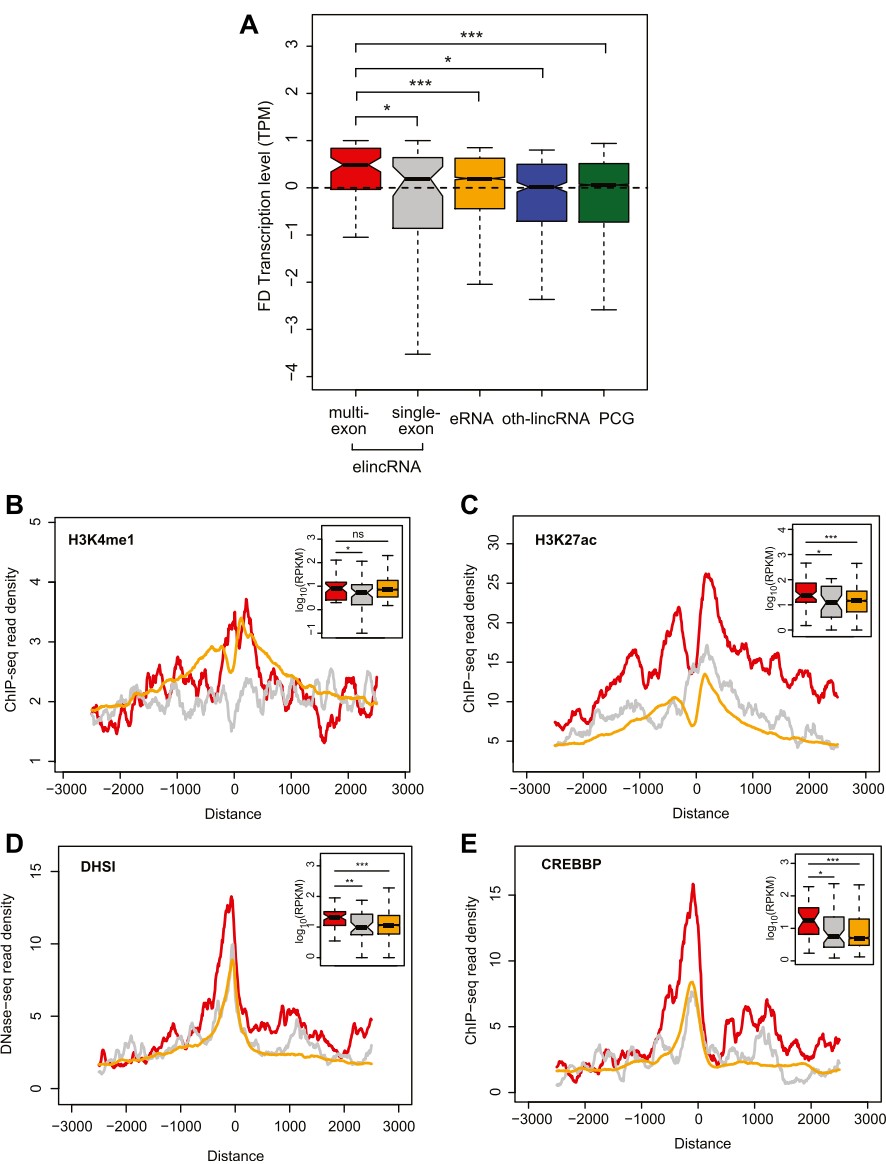

**Figure 2.   Multi-exonic elincRNAs are associated with higher enhancer activity.**
**(A)** Distribution of the fold difference (FD) in transcription (measured as CAGE TPM) of the most proximal gene to multi-exonic (red) and single-exonic (grey) elincRNAs, eRNAs (yellow), other mouse embryonic stem cell–expressed lincRNAs (oth-lincRNAs, blue), and protein-coding genes (PCGs, green) both expressed in a same stage of embryonic neurogenesis. Fold difference of neighboring genes is calculated between the two cellular stages across neuronal differentiation, where the expression level of their reference locus (elincRNA, oth-lincRNA, or PCG) is maximal and minimal. **(B, C, D, E)** Metagene plots and distribution (figure insets) of (B) H3K4me1, (C) H3K27ac, (D) DNase I hypersensitive sites (DHSI), and (E) Crebbp ChIP-seq reads in mouse embryonic stem cells at transcription-initiation regions of multi-exonic (red) and single-exonic (grey) elincRNAs and eRNAs (yellow). Differences between groups were tested using a two-tailed Mann–Whitney *U* test. *$P < 0.05$; ***$P < 0.001$.

multi-exonic elincRNAs are more likely to physically interact with CREBBP than are other enhancer-derived RNAs.

## Multi-exonic elincRNAs are specifically associated with changes in local chromosomal architecture

Because *cis*-regulatory interactions are dependent on local chromosomal architecture, we examined whether the observed association between elincRNA splicing and enhanced neighboring gene expression was mediated through the modulation of their local chromosomal organization.

Analysis of their relative position within mESC TADs revealed that only multi-exonic elincRNA TIRs were significantly enriched at TAD boundaries and depleted at TAD centers ($P < 0.05$, Fig 3A). This suggests that elincRNAs' preferential location at TAD boundaries (Tan et al, 2017) is restricted to multi-exonic elincRNAs. Preferential

localization of multi-exonic elincRNA-transcribing enhancers at TAD boundaries, where chromosomal looping between enhancers and promoters frequently occurs (Symmons et al, 2014; Lupianez et al, 2015), is further supported by the enriched binding of protein factors implicated in the establishment and modulation of chromosomal topology (Bonev & Cavalli, 2016). Relative to their single-exonic counterparts, multi-exonic elincRNA-producing enhancers display evidence for higher binding of Ctcf (Fig S2A), subunits of the cohesin complex (Smc1a and Smc3), its cofactor Nipbl (Fig S2B–D), and the mediator complex (Med1 and Med3) (Fig S2E and F) in mESCs.

Enhancer-associated transcripts participate in enhancer-promoter looping by recruiting Cohesin or Mediator complexes to enhancer regions, which in turn stimulate cognate target gene transcription (Lai et al, 2013; Hsieh et al, 2014). Consistent with the role of multi-exonic elincRNAs and their underlying enhancers in cell-type–specific

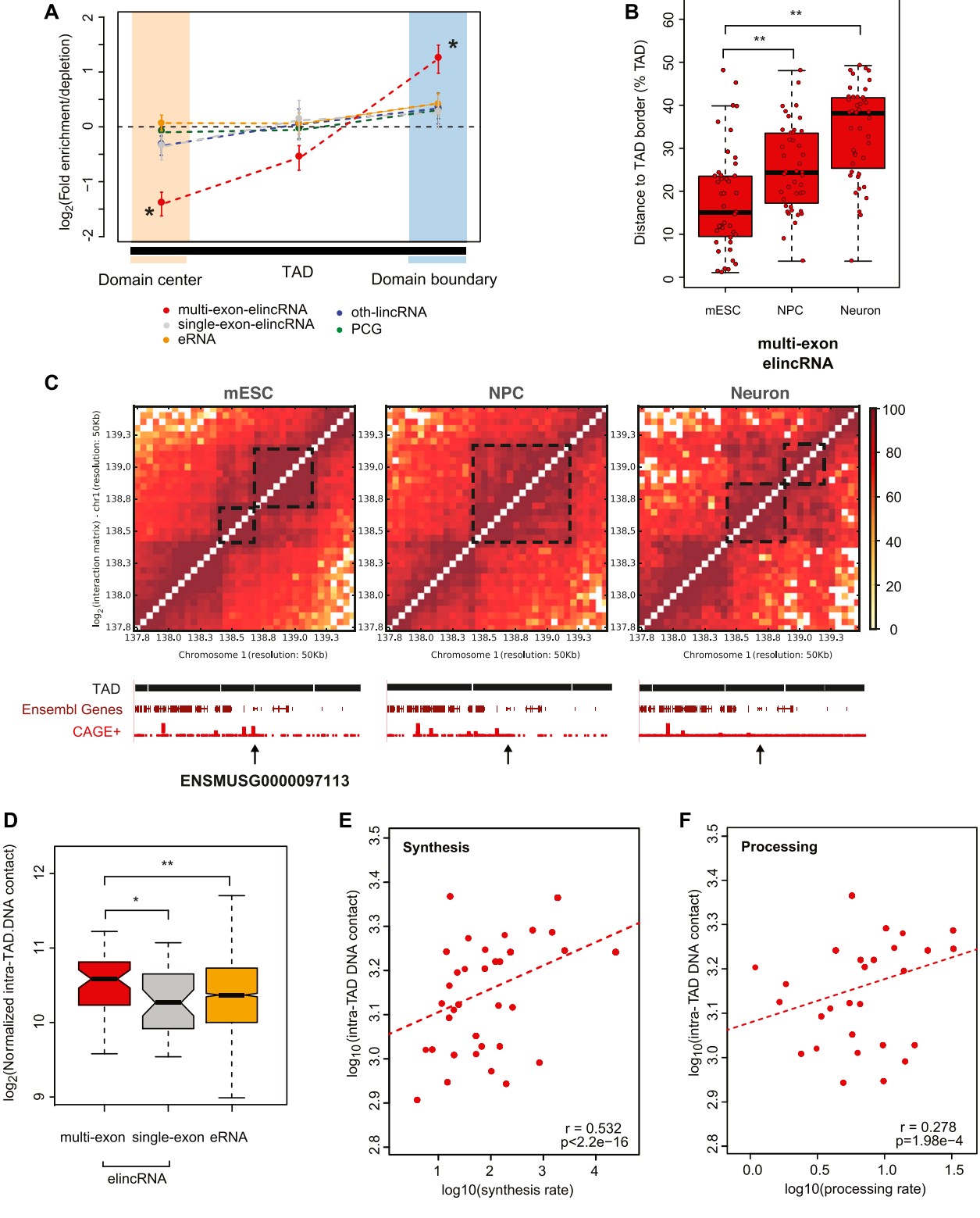

**Figure 3. Multi-exonic elincRNAs are associated with modulation of local chromosomal architecture.**
**(A)** Fold enrichment or depletion of multi-exonic (red) and single-exonic (grey) elincRNAs, eRNAs (yellow), other expressed lincRNAs (blue), and protein-coding genes (green) at boundaries (light blue shaded area) and center (light yellow shaded areas) of TADs. Significant fold differences are denoted with * ($P < 0.05$, permutation test) and standard deviation is shown with error bars. **(B)** Distribution of the distance between multi-exonic elincRNA transcription-initiation sites (red) to the nearest TAD border in mouse embryonic stem cells (mESCs), neuronal precursor cells (NPCs), and neurons. **(C)** Heat map displaying the amount of chromosomal interactions, measured using Hi-C data, at regions surrounding one multi-exonic elincRNA (ENSMUSG0000097113) in mESC, NPC, and Neuron. Dotted black squares denote TAD, which is

modulation of local chromosomal structure, we found that although, on average, the location of single-exonic enhancer-derived lincRNAs and eRNAs remained relatively unchanged with respect to their nearest TAD border (Fig S3A), the distance between TAD borders and multi-exonic elincRNA TIRs increases upon cell differentiation (Fig 3B and C). Multi-exonic elincRNA transcription is strongly correlated with the presence and maintenance of TAD boundaries across differentiation, supporting cell-type–specific functions of these enhancers (Fig S3B and C). Furthermore, supporting the tissue-specific activity and functions of multi-exonic elincRNA-transcribing enhancers, we found that genes in their vicinity are enriched in genes involved in mESC pluripotency maintenance (1.73-fold enrichment, $P < 0.05$, hypergeometric test) (Xu et al, 2013) and DNA binding and RNA transcription (Fig S3D).

To assess the impact of multi-exonic elincRNA on local chromosomal architecture, we next investigated the relationship between enhancer transcription and splicing and intra-TAD DNA contact density. We found that the frequency of DNA contacts within TADs that encompass multi-exonic elincRNA loci to be significantly higher than those containing other transcribed enhancers ($P < 0.05$, two-tailed Mann–Whitney $U$ test, Fig 3D, see the Materials and Methods section). Furthermore, we found that the density of local chromosomal interactions correlated with the rate of transcription (Fig 3E) and processing (Fig 3F) of multi-exonic elincRNAs.

### Activity of enhancers that transcribe multi-exonic elincRNAs is conserved

We reasoned that if splicing of enhancer-associated transcripts is biologically relevant, multi-exonic elincRNA-producing enhancers should be conserved during evolution. To test this hypothesis, we assessed the extent of enhancer conservation by overlapping the syntenic regions of transcribed mESC enhancers in humans with H1 ESC (hESC) enhancers (The ENCODE Project Consortium, 2012). We found that more than half (n = 57/100, 57%) of mESC enhancers that produce elincRNAs have conserved chromatin signatures at their syntenic regions in hESCs, a significantly higher proportion than those that produce eRNAs (n = 487/2,117, 23%, $P < 5 \times 10^{-13}$, two-tailed Fisher's exact test). Furthermore, relative to enhancers that transcribe single-exonic elincRNAs, those that express multi-exonic elincRNAs are twofold enriched among conserved enhancers ($P < 1 \times 10^{-4}$, two-tailed Fisher's exact test). Importantly, of the conserved enhancers with evidence of transcription in humans (n = 12/57, 21%), most give rise to multi-exonic elincRNAs in mESCs (n = 10/12, 83%), consistent with the conservation of the function and transcription of these enhancers during mammalian evolution.

### Rapid elincRNA splicing is associated with efficient transcription

We next turned our attention to the mechanisms and sequences underlying the splicing of elincRNAs. Differences in GC content between intronic and exonic sequences are known to facilitate splice site recognition and increase splicing efficiency (Amit et al, 2012). The exons and introns of elincRNAs display distinct GC contents, similar to protein-coding genes and oth-lincRNAs (Fig 4A) (Schuler et al, 2014; Haerty & Ponting, 2015). Further supporting the biological relevance of elincRNA splicing, we found that their splice site (SS)–flanking regions are enriched in splicing-associated elements, including exonic splicing enhancers (Fig 4B) and U1 snRNP-binding motifs (Fig 4C). Relative to other multi-exonic lincRNAs, elincRNAs SSs also have a higher likelihood of being recognized by the splicing machinery (Fig S4A and B). Together, these results suggest elincRNA splicing is efficient.

To assess whether increased density of splicing-associated motifs at multi-exonic elincRNA reflect efficient transcript splicing at these loci, we determined their transcriptome-wide rates of splicing in mESCs. We performed 4-thiouridine (4sU) metabolic labeling of RNA for 15, 30, and 60 min. Ribo-depleted total RNA from the total and newly transcribed fractions was sequenced and used to estimate transcriptome-wide rates of synthesis, splicing, and degradation in mESCs using INSPEcT (de Pretis et al, 2015) (Fig S4C). Consistent with previous reports, lincRNAs as a class were significantly less efficiently spliced than protein-coding genes (Mele et al, 2017; Mukherjee et al, 2017). However, compared with other lincRNAs, those transcribed from enhancers were 1.5-fold more rapidly processed (Fig 4D) and a higher proportion of their introns (14%) have undergone complete splicing (Fig 4E, $P < 0.05$, two-tailed Mann–Whitney $U$ test, Table S3). The splicing efficiency of elincRNAs was comparable with that of protein-coding genes (Fig 4D and E). No significant differences were found in the synthesis and degradation rates between elincRNAs and other lincRNAs ($P > 0.05$ two-tailed Mann–Whitney $U$ test, Fig S4D).

We found the exons of multi-exonic elincRNA evolved neutrally (Fig S5), suggesting efficient splicing of these transcripts was not maintained to preserve the assembly of evolutionarily conserved and likely functional sequence motifs within their primary transcripts. Given the well-established coupling between splicing and transcription (Brinster et al, 1988; Le Hir et al, 2003) and higher splicing efficiency of elincRNA 5′ exons (Fig 5A, $P < 0.05$, two-tailed Mann–Whitney $U$ test), which was not detected for mRNAs or oth-lincRNAs (Fig 5A), we questioned if splicing was instead associated with higher transcription of multi-exonic elincRNA loci. Consistent with this hypothesis, we found multi-exonic elincRNA transcripts were more rapidly synthesized than their single-exonic counterparts (Fig 5B). This higher transcriptional activity was further supported by elevated levels of engaged RNA Polymerase II (RNAPII, Fig 5C) at their TIRs and lower RNAPII promoter-proximal stalling relative to other noncoding transcripts, as shown by their relatively low ratio between RNAPII reads mapping to their TIR relative to their gene body (Travelling Ratio, Fig 5D, $P < 0.05$, two-tailed Mann–Whitney $U$ test, see the Materials and Methods section). Furthermore, relative to other non-spliced ncRNAs, multi-exonic elincRNA TIRs and gene bodies were enriched in phosphorylated serine 5 (S5P) and serine 2 (S2P) (Fig 5E and F) at

also represented by the black bars below the heat map. Gene browser view of the corresponding region displaying Ensembl gene models (dark red lines) and CAGE read density (red lines) at each cell stage. **(D)** Distribution of the average amount of chromosomal contacts within mESC TADs that contain multi-exonic (red) and single-exonic (grey) elincRNAs and eRNAs (yellow). **(E, F)** DNA–DNA contacts within multi-exonic elincRNA-containing mESC TADs (log10, y-axis) as a function of their respective (E) synthesis rate or (F) processing rate (log10, red points, Spearman's correlation). Differences between groups were tested using a two-tailed Mann–Whitney $U$ test. *$P <$ 0.05; **$P < 0.01$; ***$P < 0.001$; NS $P > 0.05$.

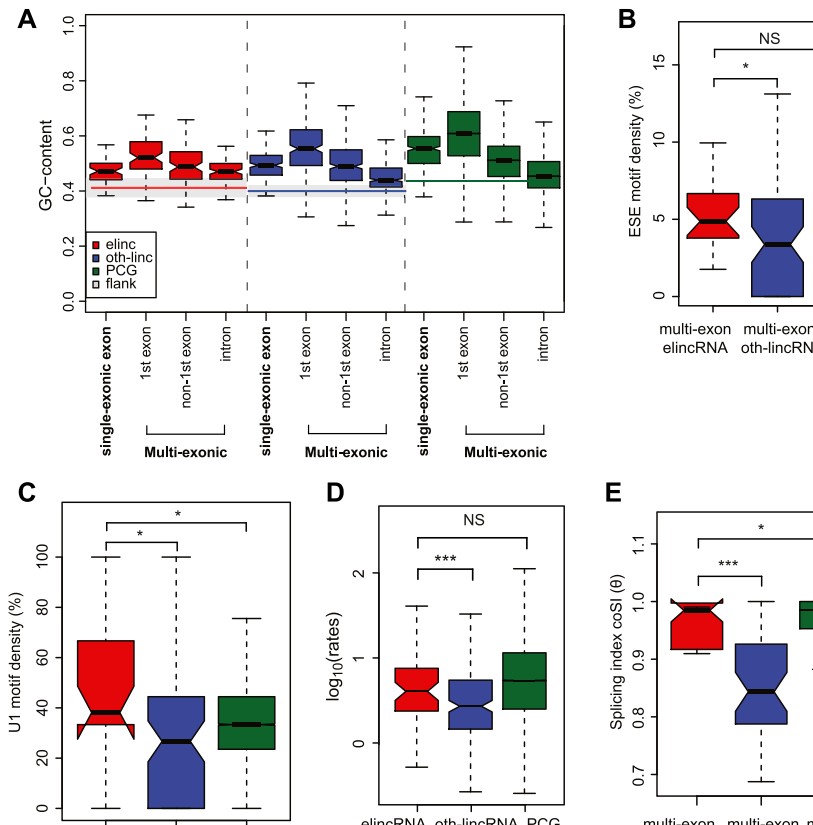

**Figure 4. elincRNA splicing is efficient.**
**(A)** Distribution of the GC content of exons and introns of single- and multi-exonic elincRNAs (red), other expressed lincRNAs (blue), protein-coding genes (green), and their respective flanking regions (grey). **(B, C)** Distribution of the density of predicted (B) exonic splicing enhancers (ESEs) and (C) U1 spliceosome RNAs (snRNPs) within multi-exonic elincRNAs (red), other expressed lincRNAs (blue), and protein-coding genes (green). **(D)** Distribution of the average processing rates for elincRNAs (red), other expressed lincRNAs (blue), and protein-coding genes (green). **(E)** Distribution of the splicing index, coSI (θ) for multi-exonic elincRNAs (red), other expressed lincRNAs (blue), and protein-coding genes (green). Differences between groups were tested using a two-tailed Mann–Whitney U test. *P < 0.05; ***P < 0.001; NS P > 0.05.

RNAPII C-terminal domain, respectively, further supporting their high transcription initiation (Ho & Shuman, 1999), efficient transcription elongation, and co-transcriptional splicing (Komarnitsky et al, 2000; Gu et al, 2013).

## Discussion

Although most active enhancers show no preference in the direction of transcription initiation or elongation and produce short and unstable eRNAs bidirectionally (Andersson et al, 2014), a fraction is expressed predominantly in one direction and give rise to elincRNAs that can be spliced (Marques et al, 2013; Hon et al, 2017). Whether differences in the directionality and transcript structure of enhancer-associated transcription underlie differences in enhancer activity remains unknown. Here, we address this question and provide evidence that enhancer-associated transcript splicing directly impact cognate enhancer function. Specifically, we found that elincRNAs, particularly those that undergo splicing, are transcribed from enhancers whose activity was conserved during mammalian evolution and are highly active. The association between elincRNA splicing and cognate enhancer activity is supported by their enrichment in enhancer epigenetic signatures; greater fold increase in putative *cis*-target expression; and the modulation of local chromosomal architecture. Our results in mouse are also consistent with recent work in human cells, which also supports that multi-exonic lincRNAs are often

transcribed from highly active enhancers (Gil & Ulitsky, 2018). Given the paucity of evidence supporting a sequence-dependent mechanism for most elincRNAs and their poor exonic nucleotide conservation, unexpectedly, we found splicing of elincRNAs is efficient.

The coupling between splicing and transcription at multi-exonic elincRNAs, particularly those at promoter-proximal exons, is also consistent with the well-established synergy between splicing and transcription (Furger et al, 2002; Damgaard et al, 2008). Our results expand on these earlier findings and reveal a novel link between elincRNA splicing and enhancer activity that in turn impact target expression. We propose that higher enhancer transcription facilitates the binding of molecular factors, such as CREBBP, the Cohesin and Mediator complexes, at their cognate enhancers, which were recently shown to induce local chromatin remodeling and conformation in an RNA-dependent manner (Lai et al, 2013; Hsieh et al, 2014; Bose et al, 2017), ultimately leading to the stronger enhancer activity observed at these loci (Fig 6).

We further propose that some enhancers associated with eRNA transcription (Andersson et al, 2014), which generally turn over rapidly during mammalian evolution (Villar et al, 2015), have evolved molecular features, including splicing that strengthened their transcription and led to increased cognate enhancer activity by facilitating the recruitment of enhancer factors in a RNA-dependent manner (Fig 6). This is in concordance with evidence that novel exon-containing transcript isoforms show increased expression (Merkin et al, 2015) and that the acquisition

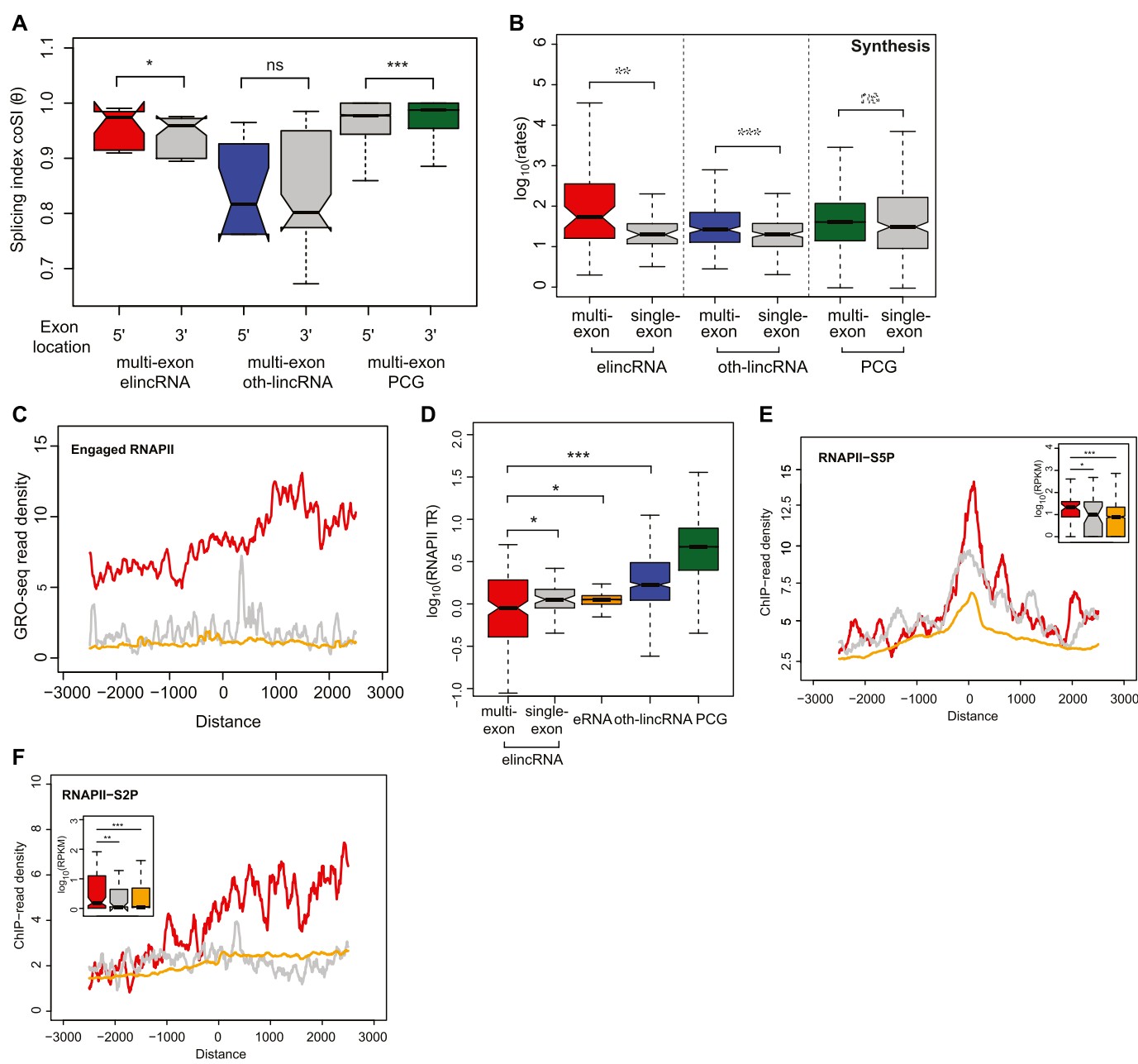

**Figure 5. elincRNA 5′ end exon splicing associates with increased transcription.**
**(A)** Distribution of the splicing index, coSI (θ) of introns located at the 5′ or 3′ ends of multi-exonic elincRNAs (red), other expressed lincRNAs (blue), and protein-coding genes (green). **(B)** Distribution of the RNA synthesis rates of multi-exonic elincRNAs (red), other expressed lincRNAs (blue), and protein-coding genes (green), as well as their single-exonic counterparts (grey). **(C)** Metagene plot of mouse embryonic stem cells GRO-seq reads centered at transcription initiation region of multi-exonic (red) and single-exonic (grey) elincRNAs and eRNAs (yellow). **(D)** Distribution of RNAPII travelling ratio (TR) for multi-exonic (red) and single-exonic (grey) elincRNAs, eRNAs (yellow), other expressed lincRNAs (blue), and protein-coding genes (green). **(E, F)** Metagene plots and distribution (figure insets) of ChIP-seq reads for RNAPII with (E) phosphorylated serine 5 (S5P) and (F) phosphorylated serine 2 (S2P) at their C-terminal domain centered at transcription initiation regions of multi-exonic (red) and single-exonic (grey) elincRNAs and eRNAs (yellow). Differences between groups were tested using a two-tailed Mann–Whitney U test. *P < 0.05; **P < 0.01; ***P < 0.001; NS P > 0.05.

of splicing and polyadenylation signals at newly evolved transcriptional initiation sites, which are intrinsically bidirectional (Jin et al, 2017), can favor the preservation of the preferred transcription direction (Almada et al, 2013; Carelli et al, 2018).

Further work is now required to establish the mechanisms underlying the evolution of efficient splicing of elincRNAs and how the processing of these transcripts facilitates recruitment of enhancer factors. Furthermore, inhibition or enhancement of splicing can be achieved through targeted approaches, such as using small molecules or antisense oligos (Spitali & Aartsma-Rus, 2012). Our results open new avenues for modulating enhancer activity through targeting elincRNA processing.

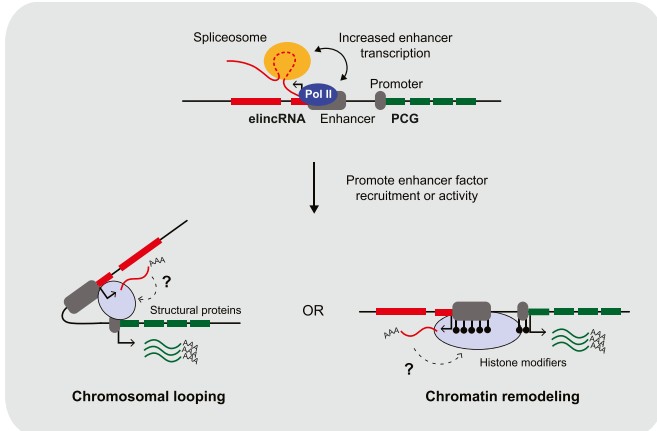

**Figure 6. Proposed model of how elincRNA splicing strengthens enhancer activity through chromatin remodeling.**
(Top panel) Enhancers (large grey box) can be transcribed by RNA Polymerase II (Pol II, blue circle) and give rise to multi-exonic elincRNA (red boxes) transcripts (red line) whose introns (dashed red line) are co-transcriptionally spliced (spliceosome, yellow circle). The synergistic interaction between elincRNA splicing and Pol II activity increases enhancer transcription, which in turn strengthens *cis*-regulation of nearby protein-coding gene targets (PCG, green, promoter as small grey box). (Bottom panel) Increased elincRNA transcription promotes RNA-dependent recruitment or activity of enhancer factors, for example: (left) structural proteins (blue shaded circle) or and (right) histone modifiers (blue shaded oval). The mechanism by which multi-exonic elincRNAs interact with enhancer factors remains unknown (question mark).

# Materials and Methods

## Identification of enhancer-associated transcripts

We considered mESC ENCODE intergenic enhancers (61,877 mESCs enhancers) (Bogu et al, 2015) to be transcribed if they overlapped DNase I–hypersensitive sites (Mouse ENCODE Consortium et al, 2012) and a cap anaysis gene expression (CAGE) cluster (Fraser et al, 2015) in the corresponding cell type (n = 2,217). We considered all mESC-expressed lincRNAs (Tan et al, 2015) and Ensembl-annotated protein-coding genes (version 70) with at least one CAGE read overlapping (by > 1 nucleotide) their first exon and an mESC CAGE cluster on the same strand. One hundred transcribed enhancers overlapped lincRNA CAGE clusters (Table S1). The remaining CAGE clusters were TIRs associated with 13,143 protein-coding genes and 317 other non–enhancer-associated mESC-expressed lincRNAs (oth-lincRNAs).

Metagene profiles of CAGE reads centered at mESC enhancers and gene TIRs were plotted using NGSplot (Shen et al, 2014). Sense and antisense reads denote those that map to the same or opposite strand, respectively, as the direction of their cognate CAGE clusters. For eRNAs, direction is defined as the direction with the highest number of CAGE clusters. In cases of equal CAGE clusters on either direction, enhancer direction is randomly assigned.

eRNAs were included only in analyses that do not require transcript models because eRNAs, by definition, are non-polyadenylated, unspliced, and shorted-lived (Kim et al, 2010).

We annotated a larger set of elincRNAs using a more permissive criterion by considering all mESC lincRNAs whose 5′ end is within 500 bp of an enhancer to be enhancer-derived. Using this approach, we identified 211 multi-exonic and 1,772 single-exonic elincRNAs. Corresponding figures for the analysis of this more comprehensive yet less stringent set of elincRNAs can be found in the Supplemental Data 1.

## Metagene analysis of binding enrichment at elincRNAs

Enrichment of histone modifications, transcription factor binding, and gene expression levels were assessed using publicly available mESC ChIP-seq and RNA-seq data sets. Downloaded data sets are listed in Table S4.

For all downloaded data sets, adaptor sequences were first removed from sequencing reads with Trimmomatic (version 0.33) (Bolger et al, 2014) and then aligned to the mouse reference genome (mm9) using HISAT2 (version 2.0.2) (Kim et al, 2015).

Metagene profiles of sequencing reads centered at gene TIRs were visualized using HOMER v4.7 (Heinz et al, 2010).

## Analysis of preferential location and chromosomal contact within TADs

mESC TADs (Fraser et al, 2015) were divided into five equal size segments where the two most external bins on either side of the TAD were considered as TAD boundaries and the middle bin as the center of TAD. Enrichment or depletion of enhancer-associated transcripts was estimated for each TAD region, relative to the expectation, using the Genome Association Tester (Heger et al, 2013). Specifically, TAD positional enrichment was compared with a null distribution obtained by randomly sampling 10,000 times (with replacement) the segments of the same length and matching the GC content as the tested loci within mappable intergenic regions of TADs (as predicted by ENCODE [Hoffman et al, 2013]). To control for potential confounding variables that correlate with the GC content, such as gene density, the genome was divided into segments of 10 kb and assigned to eight isochore bins in the enrichment analysis. The frequency of chromosomal interactions within TADs was calculated using mESCs Hi-C contact matrices (Fraser et al, 2015), as previously described (Tan et al, 2017).

## Enhancer activity across embryonic neurogenesis

Level of gene transcription initiation (CAGE-based TPM (transcripts per kilobase million) at TIRs) at each of the three stages of neuronal differentiation (mESC to NPC to neuron) was downloaded from Fraser et al (2015). Each locus was paired with its genomically closest protein-coding gene, considered here as its putative *cis*-target. Only pairs where both loci were expressed in at least one embryonic neurogenesis stage were considered. For each gene, the two stages where the locus of interest was most highly or lowly expressed were determined and used to calculate the fold difference between the expression difference of its putative *cis*-target, as described previously (Marques et al, 2013).

## Prediction of enriched transcription factor motifs at mESC enhancers

We predicted DNA motifs for transcription factors enriched at multi-exonic elincRNA TIRs (±500 bp from the center of TIRs) relative to those that transcribe single-exonic elincRNAs and eRNAs. Enrichment of motifs

of at least 8mer was predicted using FIMO (Grant et al, 2011). Enriched motifs matching with known transcription factor–binding sites (JASPAR 2016 [Mathelier et al, 2016]) were predicted using TOMTOM (Gupta et al, 2007) with default parameters.

## Expression conservation analysis

Syntenic regions of mESC (mm9) genetic elements in human (hg19) were determined using liftOver with the following parameters: -minMatch = 0.2, -minBlocks = 0.01 (Meyer et al, 2013). Regions within the ENCODE Data Analysis Consortium Blacklisted Regions (Hoffman et al, 2013) were excluded from this analysis.

We considered all transcribed mESC ENCODE intergenic enhancers (Bogu et al, 2015) to be conserved in enhancer activity if their syntenic region overlaps human ESC H1 (hESC) ENCODE enhancers (Bogu et al, 2015) by one or more base pairs. Conservation of elincRNA transcription and splicing at syntenic mESC enhancers in humans was assessed using hESC CAGE (Hon et al, 2017) and PolyA-selected RNA-seq (The ENCODE Project Consortium, 2012) data. Conserved hESC enhancers that overlapped an hESC CAGE cluster and RNA-seq reads were considered to be conserved in transcription. Those that overlapped RNA-seq reads that span across exon–intron junctions were considered to be conserved in splicing.

## 4sU metabolic labeling of mESCs and RNA extraction

Mouse DTCM23/49 XY mESCs were cultured at 37°C with 5% $CO_2$ in Knockout DMEM (#10829-018; Thermo Fisher Scientific) supplemented with 15% FBS (#16000-044; Thermo Fisher Scientific), 1% antibiotic penicillin/streptomycin (15070063; Thermo Fisher Scientific), 0.01% recombinant mouse leukemia inhibitory factor protein (#ESG1107; Merck), and 0.06 mM 2-mercaptoethanol (#31350-010; Thermo Fisher Scientific), on 0.1% gelatin-coated cell culture dishes. When confluent, the culture was divided into two and passaged eight times. Five million mESCs of two biological replicates were seeded and allowed to grow to 70–80% confluency (~1 d). RNA was labeled with 4sU (T4509; Sigma-Aldrich) and nascent RNA was isolated after the general procedure as previously described (Dolken et al, 2008). Specifically, 4sU was added to the growth medium (final concentration of 200 $\mu M$), and the cells were incubated at 37°C for 15, 30, or 60 min. The plates were washed once with 1× PBS and RNA was extracted using TRIzol (#15596-026; Thermo Fisher Scientific). 100 $\mu g$ of extracted RNA was incubated for 2 h at room temperature with rotation in 1/10 volume of 10× biotinylation buffer (Tris–HCl pH 7.4, 10 mM EDTA) and 2/10 volume of biotin-HPDP (1 mg/ml in dimethylformamide [#21341; Thermo Fisher Scientific]). RNA was extracted using phenol:chloroform:isoamyl alcohol (P3803-400ML; Sigma-Aldrich). Equal volume of biotinylated RNA and prewashed Dynabeads MyOne Streptavidin T1 beads (#65601; Thermo Fischer Scientific) was added to 2× B&W buffer (10 mM Tris–HCl, pH 7.5, 1 mM EDTA, and 2M NaCl [#65601; Thermo Fisher Scientific]) and incubated at room temperature for 15 min under rotation. The beads were then separated from the mixture using DynaMag-2 Magnet (#12321D; Thermo Fisher Scientific). After removing the supernatant, the beads were washed with 1× B&W three times. Biotinylated RNA was recovered from the supernatant after 1 min of incubation with RLT buffer (RNeasy kit, #74104; QIAGEN) and purified using the RNeasy kit according to the manufacturer's instructions.

## RNA sequencing, mapping, and quantification of metabolic rates

Total RNA libraries were prepared from 10 ng of DNase-treated total and newly transcribed RNA using Ovation RNA-seq and sequenced on Illumina HiSeq 2500 (average of 50 million reads per library).

Hundred-nucleotide-long single-end reads were first mapped to mouse ribosomal RNA (rRNA) sequences with STAR v2.5.0 (Dobin et al, 2013). On average, 20% of reads were mapped to rRNA reads. Reads that do not map to rRNA (36 million on average) were then aligned to intronic and exonic sequences using STAR and quantified using RSEM (Li & Dewey, 2011). Principal component analysis of read counts was performed to demonstrate separation between newly transcribed (labeled) and total RNA (Fig S1D). Rates of synthesis, processing, and degradation were independently inferred using biological duplicates at each labeling points using the INSPEcT Bioconductor package v1.8.0 (de Pretis et al, 2015). Biotype differences in the average rate across the three labeling times were used in the analyses (Table S2).

## GC composition

Only mESC genes with multi-exonic transcripts (two or more exons) were considered for this analysis. We computed GC content separately for the first and all remaining exons, as well as the introns, for each gene and their flanking intergenic sequences of the same length, after excluding the 500 nucleotides immediately adjacent to annotations, as previously described (Haerty & Ponting, 2015).

## Identification of splicing-associated motifs

We predicted the density of mouse exonic splicing enhancer motifs (identified in Fairbrother et al (2002)) within mESC transcripts, as described previously (Haerty & Ponting, 2015). Exonic nucleotides (50 nt) flanking the SSs of internal transcript exons (>100 nt) were considered in the analysis, after masking the 5 nt immediately adjacent to SS to avoid SS-associated nucleotide composition bias (Fairbrother et al, 2002; Yeo & Burge, 2004). Canonical U1 sites (GGUAAG, GGUGAG, and GUGAGU) adjacent to 5' SSs (three exonic nt and six intronic nt flanking the 5' SS) were predicted as previously described (Almada et al, 2013). FIMO (Grant et al, 2011) was used to search for perfect hexamer matches within these sequences. For each exon, we estimated the SS strength using MaxENT (Yeo & Burge, 2004). SS scores were calculated using the −3 exonic nt to +6 intronic nt and −20 intronic nt to +3 exonic nt flanking the 5' SS and 3' SS, respectively.

## Splicing efficiency

The efficiency of splicing was assessed by estimating the fraction of transcripts for each gene where its introns were fully excised using bam2ssj (Pervouchine et al, 2013). The splicing index, coSI ($\theta$), represents the ratio of total RNA-seq reads spanning exon–exon splice junctions (excised intron) over those that overlap exon–intron junctions (incomplete excision) (Tilgner et al, 2012).

### RNAPII stalling

Distribution of RNAPII across the gene TIR and body, commonly used as an indicator of promoter-proximal RNAPII stalling and efficient transcription elongation, was estimated by calculating the travelling ratio and by using mESC RNAPII ChIP-seq data (Brookes et al, 2012). The travelling ratio represents relative read density at gene TIRs divided by that across the gene body (Reppas et al, 2006).

### Statistical tests

All statistical analyses were performed using the R software environment for statistical computing and graphics (R Development Core Team, 2008).

### Data access

The raw and processed 4sU sequencing data generated in this study have been submitted to the NCBI Gene Expression Omnibus under accession number GSE111951. Most analyses were performed using standard publicly available command-line tools, as detailed in the Materials and Methods section.

## Supplementary Information

## Acknowledgements

We thank Chris P Ponting and the members of the Marques group for valuable comments and discussion. We thank Francesco Nicassio and Matteo Marzi for their help in establishing 4sU labeling, and Mattia Pelizzola and Stefano de Pretis for advice on RNA metabolic rate inference. This work is funded by the Swiss National Science Foundation grant (PP00P3_150667 to AC Marques) and the Swiss National Center of Competence in Research (NCCR) RNA & disease. RS Young acknowledges the support of the UK Medical Research Council (U127597124) and the Medical Research Foundation.

### Author Contributions

JY Tan: conceptualization, data curation, formal analysis, supervision, funding acquisition, investigation, methodology, project administration, and writing—original draft, review, and editing.
A Biasini: conceptualization, data curation, formal analysis, investigation, methodology, and writing—original draft, review, and editing.
RS Young: formal analysis and writing—review and editing.
AC Marques: conceptualization, formal analysis, supervision, funding acquisition, project administration, and writing—original draft, review, and editing.

### Conflict of Interest Statement

The authors declare that they have no conflict of interest.

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
