## [Reviewer comments · Life Science Alliance]

Life Science Alliance

Splicing of enhancer-associated lincRNAs contributes to enhancer activity

Jennifer Tan, Adriano Biasini, Robert Young, and Ana Claudia Marques

DOI: <https://doi.org/10.26508/lsa.202000663>

Corresponding author(s): Ana Claudia Marques, UNIL University de Lausanne

Review Timeline:	Submission Date:	2020-01-30
	Editorial Decision:	2020-01-30
	Revision Received:	2020-02-12
	Accepted:	2020-02-13

Scientific Editor: Andrea Leibfried

Transaction Report:

Please note that the manuscript was previously reviewed at another journal and the reports were taken into account in the decision-making process at Life Science Alliance. Since the original reviews are not subject to Life Science Alliance's transparent review process policy, the reports and author response cannot be published.

January 30, 2020

RE: Life Science Alliance Manuscript #LSA-2020-00663-T

Dr. Ana Claudia Marques
UNIL University de Lausanne
Faculty of Biology and Medicine
Génopode
Lausanne CH-1015
Switzerland

Dear Dr. Marques,

Thank you for transferring your revised manuscript entitled "Splicing of enhancer-associated lincRNAs contributes to enhancer activity" to Life Science Alliance. Your manuscript was reviewed twice at another journal, and the editors transferred those reports to us with your permission.

The reviewers who evaluated your manuscript at the other journal had a mixed view on your work. Two reviewers were concerned about the broader conceptual advance and the generality of the conclusions when only looking at a subset of ncRNAs. There were also concerns regarding the correlations drawn and the effect size. You already included a more comprehensive analysis using a less stringent approach to include more ncRNAs, thus addressing one of the main concerns of the reviewers. You also responded to the other concerns raised.

We discussed your work in light of the reviewer input and your responses and realize that it is a challenging field and analyses. We overall concluded that there are no technical aspects precluding publication here, and we appreciate the aim of your study. We thus would like to offer publication in Life Science Alliance, and we invite you to submit a final manuscript version to us.

When uploading the revised version of your manuscript, please pay attention to the following:

- Because of the changes still to be introduced, we have not carefully checked whether all figures are called out within the manuscript text, please make sure that you do; note that the supplementary code is currently not called out
- Please include the in-house software and R code used to mine and analyze the results or deposit them in a repository
- Please upload the manuscript text in docx format
- Please make sure to upload all figure files as individual files (also supplementary figures); the legends (including supplementary figure and supplementary table legends) should be within in the main manuscript docx file
- Please incorporate the suppl methods into the main manuscript text
- Please enter all information in our submission system, such as author contributions, key words, running title, and summary blurb
- Please make sure that all corresponding authors link their profile in our system to their ORCID iD

A. FINAL FILES:

B. MANUSCRIPT ORGANIZATION AND FORMATTING:

****Reviews, decision letters, and point-by-point responses associated with peer-review at Life Science Alliance will be published online, alongside the manuscript. If you do want to opt out of having the reviewer reports and your point-by-point responses displayed, please let us know**

immediately.**

Sincerely,

February 13, 2020

RE: Life Science Alliance Manuscript #LSA-2020-00663-TR

Dr. Ana Claudia Marques
UNIL University de Lausanne
Faculty of Biology and Medicine
Génopode
Lausanne CH-1015
Switzerland

Dear Dr. Marques,

Thank you for submitting your Research Article entitled "Splicing of enhancer-associated lincRNAs contributes to enhancer activity". I appreciate how you restructured your manuscript and the newly introduced figure 1, and it is a pleasure to let you know that your manuscript is now accepted for publication in Life Science Alliance. Congratulations on this interesting work.

DISTRIBUTION OF MATERIALS:

Again, congratulations on a very nice paper. I hope you found the review process to be constructive and are pleased with how the manuscript was handled editorially. We look forward to future exciting submissions from your lab.

Sincerely,

Andrea Leibfried, PhD
Executive Editor
Life Science Alliance
Meyerohofstr. 1
69117 Heidelberg, Germany
t +49 6221 8891 502
e a.leibfried@life-science-alliance.org
www.life-science-alliance.org